# Apply a Physiologically Based Pharmacokinetic Model to Promote the Development of Enrofloxacin Granules: Predict Withdrawal Interval and Toxicity Dose

**DOI:** 10.3390/antibiotics10080955

**Published:** 2021-08-08

**Authors:** Kaixiang Zhou, Aimei Liu, Wenjin Ma, Lei Sun, Kun Mi, Xiangyue Xu, Samah Attia Algharib, Shuyu Xie, Lingli Huang

**Affiliations:** 1National Reference Laboratory of Veterinary Drug Residues (HZAU) and MAO Key Laboratory for Detection of Veterinary Drug Residues, Wuhan 430070, China; flyingkai@webmail.hzau.edu.cn (K.Z.); 2018302010040@webmail.hzau.edu.cn (A.L.); mawenjin@webmail.hzau.edu.cn (W.M.); sunlei@webmail.hzau.edu.cn (L.S.); mikun@webmail.hzau.edu.cn (K.M.); xuxiangyue@webmail.hzau.edu.cn (X.X.); samah.alghareeb@fvtm.bu.edu.eg (S.A.A.); xieshuyu@mail.hzau.edu.cn (S.X.); 2Department of Clinical Pathology, Faculty of Veterinary Medicine, Benha University, Moshtohor, Toukh 13736, Egypt; 3MOA Laboratory for Risk Assessment of Quality and Safety of Livestock and Poultry Products, Huazhong Agricultural University, Wuhan 430070, China

**Keywords:** enrofloxacin, physiologically based pharmacokinetic model, residue, liver toxicity, development of antibiotic

## Abstract

Enrofloxacin (ENR) granules were developed to prevent and control the infections caused by foodborne zoonotic intestinal pathogens in our previous studies. To promote the further development of ENR granules and standardize their usage in pigs, a physiologically based pharmacokinetic (PBPK) model of the ENR granule in pigs was built to determine the withdrawal time (WT) and evaluate the toxicity to pigs. Meanwhile, the population WT was determined by a Monte Carlo analysis to guarantee pork safety. The fitting results of the model showed that the tissue residual concentrations of ENR, ciprofloxacin, and ENR plus ciprofloxacin were all well predicted by the built PBPK model (R^2^ > 0.82). When comparing with the EMA’s WT1.4 software method, the final WT (6 d) of the ENR granules in the population of pigs was well predicted. Moreover, by combining the cytotoxicity concentration (225.9 µg/mL) of ENR against pig hepatocytes, the orally safe dosage range (≤130 mg/kg b.w.) of the ENR granules to pigs was calculated based on the validated PBPK model. The well-predicted WTs and a few uses in animals proved that the PBPK model is a potential tool for promoting the judicious use of antimicrobial agents and evaluating the toxicity of the veterinary antimicrobial products.

## 1. Introduction

Intestinal bacterial infections that are caused by zoonotic pathogens, including *Campylobacter jejuni* (*Camp. jejuni*), *Salmonella* spp., and *Escherichia coli* (*E. coli*), have resulted in serious threats to human health, and enormous economic losses to livestock and poultry breeding [1,2]. To reduce the harms that are caused by these zoonotic pathogens to humans and animals, an enrofloxacin (ENR) granule, with good palatability and fast dissolution, was developed in our previous studies [3]. The ENR granule was prepared by solid dispersion technology, and starch, as well as sodium chloride, was used as the excipient. Meanwhile, the potential dose regimen of ENR granules to the infections caused by *Salmonella* and *E. coli* was proposed as 5 mg/kg b.w. once daily (Appendix A) [3]. Even if there is not an ENR oral product for pigs on the market currently (Drug Bank, Animal Drug @ FDA), the fluoroquinolones (e.g., ENR) are largely used in the veterinary clinic worldwide [4]. In view of the favorable antibacterial effect of ENR against common intestinal pathogens, the good palatability of the ENR granules for pigs, and the convenience of administrating them orally in a group, the developed ENR granules are expected to be extensively used in the veterinary clinic.

The excessive residue of veterinary drugs in animal-derived food will cause various adverse reactions and toxicity to humans, such as diarrhea, vomiting, allergies, and hormone-like reactions [5,6,7]. For a veterinary (food-producing animals) antibiotic product on the market to be successful, its withdrawal time (WT) needs to be determined. However, how long the WT in the pigs needed to be adopted after the developed ENR granules were administrated to the pigs was unknown. According to the Food and Agriculture Organization of the United Nations (FAO), pork is the world’s most consumed meat from terrestrial animals [8]. Therefore, to promote the further development of the developed ENR granules, to standardize the use of the ENR granules on pigs, and to ensure the safety of pork products for consumers, the WT of the developed ENR granules in pigs needs to be determined. Moreover, ENR belongs to a broad-spectrum antibiotic. It shows high antibacterial activity to several bacterial pathogens, for instance, *Salmonella*, *E. coli*, *and Mycoplasma* [3]. Meanwhile, due to the favorable pharmacokinetics (PK), and the convenience of the administration (mixing feed) of ENR granules, the developed ENR granules may be used label-extra by farmers, thus cause food safety and toxicity worries in humans. However, once the veterinary drugs are used label-extra, the safe WT is difficult to be determined by the conventional slaughter method. Additionally, fluoroquinolones have been categorized as critical for human use by the World Health Organization (WHO). Their excessive residue will induce bacterial resistance, which will cause major threats to public health. Therefore, it is essential to determine the WT of ENR granules, and to predict how extensive the WT is by a silico method, following its label-extra use. As a result, this would limit excessive residues in pigs, therefore reducing resistance in humans.

On the other hand, as mentioned above, there is no ENR oral product for pigs on the market currently. Therefore, the target species safety needs to be studied for further development of the ENR granules. Meanwhile, Wang et al. (2016) proved that when ENR was orally administered to *Acipenser baerii*, at a dose of 80 mg/kg b.w. for 3 days, the atrophy and apoptosis of the hepatocytes were observed [9]. Also, when pigs were orally administrated by enteric ENR granules, at a dose of 5 mg/kg b.w. twice daily for 5 days, the ENR concentration in the pigs’ tissues was higher in the liver > kidney > muscle > fat [10]. Obviously, the high concentration of ENR was toxic to hepatocytes, while the liver is probably the accumulation target tissue of ENR. The conclusions meet the reports of FAO that the highest ENR concentration initially occurred in the liver in all the experimental species. Moreover, in all the species studied, elimination was primarily via the urine and feces (FAO, http://inchem.org/documents/jecfa/jecmono/v34je05.htm, accessed on 2 November 2001). Therefore, to evaluate the toxicity of ENR granules to pigs, the potential toxicity of the developed ENR granules for pigs, especially for the liver, needs to be determined.

However, according to the guiding principles of the International Cooperation on Harmonization of Technical Requirements for the Registration of Veterinary Medicinal Products (VICH), many animals are needed for processing the target species safety and the WT experiments. Briefly, for determining the WTs and the target species safety of a veterinary drug product, at least 16 pigs (40–80 kg) and 32 pigs were needed to be slaughtered, respectively [11,12]. Undoubtedly, the large number of animals used will cause a huge economic burden to drug companies and scientific institutions, thus limit the food safety estimation and toxicity study of veterinary antimicrobial products. Meanwhile, more experimental animal uses means that there will be more animal welfare issues. Therefore, a replaceable tool that could be used to predict the residual profiles and toxicity dose, with fewer animal uses, is urgently needed.

The physiologically based pharmacokinetic (PBPK) model is a mechanistic model that could simulate the real-time dynamic processes of drugs in different organs and at different doses, by incorporating the physiological parameters of animals and the physicochemical parameters of drugs [13]. Because of its great prediction and extrapolation power, the PBPK model was widely applied to predict the tissue residues of veterinary drugs in pigs, cattle, and fish [14,15,16,17], and to predict the toxicity of compounds [18,19]. Undoubtedly, making use of this in silico alternative testing strategy to study the WTs and toxicity of drugs, fits the 3R principle.

To ensure the safety of pork products to customers, and the safety of ENR granules to pigs, the WT and the liver toxicity of the prepared ENR granules in pigs need to be determined. To reduce animal use during the WT and toxicity studies, the PBPK model of ENR in pigs was built, by combining the physiological parameters of pigs and the chemical-specific parameters of ENR. Then, the WTs of the ENR granules, after multiple oral administrations to pigs, were predicted by Monte Carlo (MC) analysis. Besides, the in vivo liver toxicity dose of the ENR granules was obtained via combining the validated PBPK model with the in vitro IC_50_ of ENR against pig hepatocytes. We emphasized that the PBPK model method showed enormous potential in reducing animal use during the development of new veterinary antibiotics, compared with the conventional WT and toxicity study methods. This is also the first time that the PBPK model has been applied to study the toxicity of veterinary antibiotics.

## 2. Results

### 2.1. Calibration and Evaluation of the PBPK Model

The built PBPK model was verified by the observed plasma concentrations and tissue residue concentrations. Because the predicted plasma concentrations were fitted by the observed concentrations in our previous studies (Appendix A) [3], the PBPK model was directly used to predict the tissue residual concentrations of ENR, ciprofloxacin (CIP), and ENR plus CIP in pigs, after they were administrated at a dose of 5 mg/kg b.w., twice per day for 5 days. Thereby, the actual residue concentrations that were obtained by the residue experiments (Appendix A) were used to fit with the predictive tissue residual concentrations. As shown in Figure 1, Figure 2, and Figure 3, the predicted residual concentrations of ENR, CIP, and ENR + CIP were well fitted by the observed concentrations. The linear regression analysis results of the observed and predicted data showed that all the determination coefficients (R^2^) were >0.82 (Appendix A). These suggested that the built PBPK model possessed high predictive accuracy.

### 2.2. Sensitivity Analysis

The parameters with an absolute value of a normalized sensitivity coefficient (NSC) ≥0.2 were considered as sensitive parameters, and they are listed in Appendix A. The results indicated that the concentration of ENR plus CIP at 228 h (final concentration) in the tissues was sensitive to BW, Frac, Kmc, Kurine1c, Kurinec, Pf, Pf1, PK, Pk1, Pl, Pl1, Pm, and Pm1. The final concentration was sensitive to Kmc, Kurine1c, and Kurinec, suggesting that the liver or kidney injury would influence the WT of the ENR granules in pigs. The final concentrations of ENR plus CIP in the muscle, fat, liver, and kidney were the most sensitive to BW with NSC values above 1.8, suggesting that the BW influenced the final concentration significantly. These indicated that the WT of ENR granules in pigs of different ages will be different, and this is a reminder to ensure the safety of pork products; the WT of ENR granules in a population of pigs that possess different physiological conditions needs to be determined. Subsequently, the MC analysis was performed, based on the distribution of sensitive parameters that were produced by the above parameter-sensitive analysis (PSA).

### 2.3. Population Withdrawal Interval Estimation

To ensure that the pork products are safe enough for humans, the population WTs of ENR granules in pigs were performed by a Monte Carlo (MC) analysis. The values and distributions of the parameters that were used in the MC analysis are listed in Appendix A. After 1000 iterations were performed, the numbers of residue concentrations of ENR plus CIP, in virtual individual values below the tissue maximum residue limit (MRLs) at different withdrawal intervals, were counted. When the residue concentrations of 99% virtual individual values were below the MRLs, the interval time after the last administration was treated as the predicted WT of each tissue. As shown in Figure 4, the predictive WT that was calculated by the MC analysis in the muscle, fat, liver, and kidney was 5 d, 3 d, 6 d, and 4 d, respectively. Meanwhile, the actual WTs of the tissues that were obtained by the EMA’s WT-1.4 software, are shown in Appendix A. The results indicated that the actual WTs of the ENR granules in the muscle, fat, liver, and kidney were 3 d, 2 d, 4 d, and 6 d, respectively. The results indicated that the predictive WTs in edible tissue were closed with the measured WTs (Appendix A). As required by VICH, the longest WT in tissues is defined as the final WT of the veterinary products [12]. Therefore, the measured WT and predicted WT of the prepared ENR granules were both 6 d.

### 2.4. Reversed IC_50_ to Orally Toxicity Dose

The viability of pig hepatocytes in the presence of ENR and CIP was obtained by in vitro inhibition experiments. When the concentration of CIP was 500 µg/mL, the viability of the pig hepatocytes was 58.86 ± 3.70%. Moreover, the viability of the cells was 13.83 ± 3.70%, after being treated with 500 µg/mL of ENR (Appendix A). This indicated that the pig hepatocytes toxicity of ENR was significantly higher than that of CIP. Besides, ENR can produce CIP through metabolizing in the liver in vivo. The ENR showed a significantly higher peak concentration (C_max_) than that of CIP in the liver, at the dose of 5 mg/kg b.w. (Figure 1 and Figure 2). As a reminder, the in vivo liver toxicity of the ENR granules was mainly caused by ENR, rather than CIP. Therefore, the liver toxicity was mainly analyzed by ENR. Subsequently, the cell growth curve was drawn and the IC_50_ value of ENR against pig hepatocytes was calculated by the growth curves. As shown in Figure 5a, the IC_50_ value of ENR was log 2.354 µg/mL (225.9 µg/mL). We assumed that when the concentration of ENR in the in vivo liver reached this IC_50_ value, the ENR granules would show liver toxicity to pigs. Because the predicted drug concentrations in the liver were well verified by the observed data, the built PBPK model could be used to extrapolate the concentration curves in the liver, at different doses. According to the predicted dose curves of ENR in the liver, when the oral dose was 130 mg/kg b.w., the C_max_ of ENR in the liver was 222.9 µg/mL (Figure 5b). As mentioned above, the in vitro IC_50_ of ENR against pig hepatocytes was 225.9 µg/mL. Therefore, when orally administrated at a dose of 130 mg/kg b.w., the in vivo hepatocytes would be exposed at a toxicity concentration. In other words, the in vivo liver safe dose range to pigs, of the prepared ENR granules, was ≤130 mg/kg b.w. Meanwhile, the time to reach the C_max_ was 109 h. At the same time, the CIP concentration was 51.9 µg/mL (Figure 5b). Due to the low inhibition rate caused by 51.9 µg/mL CIP (Appendix A), the toxicity effect of CIP against pig hepatocytes was ignored.

## 3. Discussion

In the present study, to determine the WT and evaluate the toxicity of previously developed ENR granules with a few animal uses, a PBPK model for ENR granules in pigs was established. After the constructed model was validated by the measured residual data, the WT for a population of pigs was determined by an MC analysis. Then, the toxicity dose of the ENR granules for pigs was calculated, based on the in vitro IC_50_ of ENR against hepatocytes. In other words, the WT and toxicity data that are required by a new veterinary antibiotic product, were collected by an in silico method rather than animal experiments. Furthermore, the good predictive accuracy of the constructed PBPK model showed that the application of this PBPK model can help to predict the extended withdrawal intervals once the ENR granules were used extra-label in the clinic, and simplify the target species experiments, thus ensure the safety of pork products to humans and the safety of ENR granules to pigs, therefore promoting the judicious use of the developed ENR granules on pigs.

During the model validation, because the actual data from five time points were used to fit the predicted data, the conclusion of the high predictive accuracy of the built PBPK model could be made in the present study. For instance, as reported by Yang et al. (2019), the predicted florfenicol amine concentrations in the muscle of cattle were verified by four actual data, and those in the liver were verified by five actual data [11]. Besides, according to the requirements of the VICH, for the WT determination of a veterinary drug for pigs, four sampling time points are enough. Therefore, the tissue residual concentration of ENR granules in pigs, at different doses and different ages, could be well predicted by the validated model. What is worth mentioning is the good prediction of the C_max_ in edible tissues, which provided more confidence for predicting the toxicity dose.

Compared with focusing on the final concentration in the present studies, the PSA in previous residual studies was performed, by focusing on the effects of the parameters on the area under the drug concentration–time curve (AUC) [17]. Because the WT of the ENR granule is defined as the time when the concentration of ENR plus CIP is below the MRL, the best-related value with the WT is the retention time of ENR and CIP in tissues. Although the AUC was commonly used as the target of sensitivity analyses, the high C_max_ with a short retention time might show the same AUC with the low C_max_ possessing a long retention time. Obviously, the WTs between them would be different. Therefore, after careful consideration, the PSA was performed by focusing on the effects of the parameters on the final concentration of ENR plus CIP in the present study.

In this paper, similarly to what Yang et al. (2021) reported, firstly the 1000 virtual individuals with different physiological parameters were simulated, by giving a default 10% coefficients of variance (CVs) to the sensitive parameters [20]. However, the calculated WTs in the edible tissues were significantly shorter than the measured WTs. As a reminder, the 10% CVs in pigs is not enough. Subsequently, the CVs of the sensitive parameters were changed to 20% or 30%, to describe the intraspecific differences of pigs [21,22]. Because of the more time-consuming approach, the other modeling parameters with an NSC <0.2 were not taken into consideration. In this model, the actual WTs were estimated with a tolerance limit of the 99th percentile, with a 95% confidence for each tissue [17]. Then, these values were calculated, after rounding up to the next whole day, according to WT estimation criteria [17]. Reportedly, the WT values that were calculated by the EMA’s method could keep consistent with the values, using one MC simulation [20,23]. Therefore, using one MC analysis, with 1000 iterations for each tissue, is sufficient to predict the WTs of the ENR granules in tissues. In the present study, both the final WTs of the model-predicted and the measured values are 6 days, which demonstrated the high predictive accuracy of the PBPK model, and the rationality of the determination standard of the NSC and parameter CVs. However, the PSA and MC analysis belong to statistical analysis after all. The setting of CVs is an optimization process, rather than the reality parameters of experimental pigs. Therefore, the slight WT discrepancies between the model-predicted and measured values can be accepted. For instance, the predicted WTs in grass carp tissues were longer than the measured WTs, even a two-fold difference was observed [17].

Overall, our study suggested that the PBPK models are a meaningful method in the determination of the WTs of veterinary products. Compared to the conventional EMA’s method, PBPK models are more preferred because they can be directly applied to conduct extrapolation across doses, ages, routes, and even across species, rather than conducting extra animal experiments. For instance, we can obtain the WT at a dose of 10 mg/kg b.w. or 20 mg/kg b.w. by changing the dose value of the modeling code directly, rather than conducting extra animal experiments. This is meaningful for monitoring the WT in the clinic after label-extra drug use [21]. Obviously, this is impossible for the traditional EMA’s method. Undoubtedly, the PBPK models are significant for reducing animal use in the determination of WTs, and guaranteeing food safety. It will be of benefit for simplifying the development processes of new veterinary antibiotics.

What is worth discussing is the fact that, due to the pigs being slaughtered by carotid bloodletting and the liver cells being hard to destroy by simple shear action, the measured concentration of ENR in the liver (liver homogenate fluid without blood) can be treated as the concentration of the liver interstitial fluid. Therefore, the predicted concentration of ENR in the liver, which was verified by the actual concentration, can be viewed as the concentration of liver interstitial fluid. In other words, the predicted in vivo C_max_ of the liver indicates the direct exposure concentration of ENR to hepatocytes in vivo, which was related to the in vitro viability inhibition concentration in the cell medium [24]. Additionally, as mentioned above, the in vitro viability inhibition concentration was the free-drug concentration of the cell exposure medium, which was confirmed by HPLC. Also, the ENR that was extracted from the liver tissue, by KH_2_PO_3_, was also a free drug, since the drug that was bound with protein was hard to extract. Besides, the protein–drug mixture would be precipitated after nitrogen drying. Therefore, the in vitro IC_50_ and the in vivo equivalent C_max_ were both free-drug concentrations. In other words, the viability inhibition that was caused by IC_50_ and C_max_ to hepatocytes, would not be influenced by the protein-binding differences between the in vitro cell medium and the in vivo liver tissue.

Subsequently, the doses that could produce the concentration of cell exposure medium were all reversed; thus, the range of toxicity doses of the ENR granules was clear (Figure 5b). Although making use of a validated PBPK model to extrapolate the dose curves is common [17], to predict the residual C_max_ in the liver with high accuracy and confidence, another extra time point (0.042 d) was adopted to obtain the measured residual C_max_ in the liver in the present study. According to Figure 1, Figure 2 and Figure 3, the residual C_max_ in the edible tissues was well predicted by the constructed PBPK model. In other words, the residual C_max_ in the liver can be predicted with high accuracy. Therefore, we believed that the predicted toxicity dose was reliable in the present study.

Although, the results of the residue elimination experiments showed that the ENR concentration in the kidney was close to the liver’s ENR concentration (Appendix A). However, the kidney is the excretion organ of ENR; we argued that the kidney cells were more tolerant to ENR. Moreover, the ENR concentration in the liver was significantly higher than that in the muscle and fat (*p <* 0.05). Therefore, the liver was selected as the toxicity target organ of the pig in the present study. We emphasized that the liver toxicity dose of the ENR granules was assessed without any animal use. Meanwhile, if we could obtain the IC_50_ values of ENR against other tissue cells (e.g., lung, kidney), the toxicity doses to other tissues could also be reversed by the PBPK model, thus the total target species safety data could be collected completely, without any animal slaughter. After all, the blood biochemistry and blood routine data that are required by the target species safety experiments could be obtained by sampling blood in living animals. Although adopting the PBPK model, to predict the WTs of veterinary antibiotics in edible tissues, is common, this is the first time that the toxicity dose of veterinary antibiotics has been reversed by the PBPK model. Undoubtedly, the application of the PBPK model in the development of a new veterinary antibiotic would reduce the development costs and simplify the regulation, especially for the candidate compounds. At least, whether the expensive target species safety experiments of the candidate antibiotics were worth being continued could make a decision, predictively.

Importantly, we could also evaluate the population liver toxicity to pigs at five-fold the suggested dose, on the basis of the sensitive parameters, to ensure that the ENR granules are safe to 99% of pigs. Although the high liver toxicity dose (130 mg/kg b.w.) indicated that this further evaluation was not necessary, the idea is meaningful to ensure environmental exposure antibiotics, and the potential toxins in food or water are safe enough for humans. For instance, the average daily intake of several herbs and vegetables that contain anthraquinones were estimated, by combining the liver toxicity concentration and the PBPK model [13].

## 4. Materials and Methods

### 4.1. Materials

Pig hepatocytes were isolated from pig liver. ENR (content: 99.0%) and ciprofloxacin (CIP) hydrochloride (content: 94.0%) standards were provided by Dr. Ehrenstorfer Gmbh. ENR granules (content: 5%) were prepared by the Veterinary Medicine Research Center of Huazhong Agricultural University, Wuhan, China. Phosphoric acid (grade: HPLC, 80–90%) and triethylamine (grade: HPLC) were purchased from Aladdin (Shanghai, China). Fetal bovine serum, DMEM medium, and dimethyl sulfoxide (DMSO) were provided by Sigma-Aldrich (Louis, MO, USA). CCK-8 cell counting kit was purchased from Vazyme Biotech Co., Ltd. (Wuhan, China). C18 solid-phase extraction column (HLB, 3CC 60 mg) was provided by Waters (Milford, UT, USA). KH_2_PO_4_ and NaOH were provided by Sinopharm Group Chemical Reagent Co., Ltd. (Shanghai, China).

### 4.2. Animal

Twenty clinically healthy three-way hybrid pigs (55 ± 10 kg) were provided by Jinling pig farm (Wuhan, China). The pigs were fed in laboratory animal rooms at the National Reference Laboratory of Veterinary Drug Residues (HZAU). According to the requirement of the National Research Council (US) Committee [25], they were fed with drug-free feed and drink for seven days. The environment was kept at a suitable relative humidity (45–65%) and temperature (18–25 °C), respectively. All the experimental protocols were approved by the Institutional Animal Care and Use Committee at Huazhong Agricultural University (approval ethical number: HZAUSW-2019-024, data: August 2019) and followed the guidelines of Hubei Science and Technology.

### 4.3. Tissue Residues of ENR Granules in Pigs

Twenty clinically healthy three-way hybrid pigs (55 ± 10 kg) were divided into control and experimental groups after the cleaning period. Two pigs in the control group and eighteen pigs in the experimental group. To ensure the pork products are safe enough for humans, when pigs were administered the ENR granules, the shorter administration interval was adopted in the present study. Therefore, the pigs in the experimental group were administrated ENR granules at a dose of 5 mg/kg b.w. twice per day by mixed feed. According to the daily intake of the pigs (about 3 kg/day), the feed for the experimental group was adjusted to per kilogram feed containing 180 mg ENR (3600 mg ENR granules). The pigs in the control group were given drug-free feed. Moreover, all the pigs were fed twice per day. After being administrated for five consecutive days, one pig in the control group was killed at 0.042 d and 5 d, respectively, and three pigs from the experimental group were slaughtered at 0.042, 0.5, 1, 2, 3, and 5 d, respectively.

According to the European Union (EU) Directive 2010/63/EU, to reduce the pain of pigs during slaughtering, the pigs were electrically stunned firstly by an automated holding device (Stork RMS, Lichtervoorde, The Netherlands). Briefly, electricity was applied at the following three points: two in front of the ears, at the temporal fossae (50 Hz, 1.40 A, and 350 V), and one at the heart (50 Hz, 0.60 A, and 90 V), and the stunning time was 4 s. After pigs were stunned, the carotid bloodletting method was adopted to slaughter. After slaughter, the moderate weight of the muscle, fat, kidney, and liver were collected from the dissected pigs, and the tissue samples were saved at −20 °C for further residue concentration analysis. The WTs were calculated by EMA WT1.4 software, with a tolerance limit of the 99th percentile with a 95% confidence, and they were considered as observed data for estimating the predictive accuracy of the built model.

The residue detection method of ENR in pig edible tissues was reported in other literature [10]. Briefly, the 2 g tissue homogenate was added into a 50 mL tube containing 20 mL KH_2_PO_3_ buffer, then the mixture was vortexed for 5 min to extract the ENR from the tissues. After the vortex, the mixture was centrifuged at 12,000 rpm for 20 min. Then the supernatant was treated by an HLB column (Waters, Milford, UT, USA). After being washed by 1 mL methanol and 1 mL 5% ammonia methanol solution, the ENR and CIP in the tissues were collected in a 10 mL tube. When a 2 mL solution was dried by nitrogen (40 °C), the lower sediment was re-dissolved by 1 mL of methanol to obtain the final sample. After the final sample was filtered by a 0.2 μm filter, the residue drug concentration was detected by fluorescence HPLC (Waters2475, Waters, Milford, UT, USA). A C_18_ column (SB-Aq, 250 × 4.6 mm, i.d., 5 μm, Agilent, Santa Clara, CA, USA) was used for HPLC, which was performed with an excitation wavelength of 280 nm and an emission wavelength at 450 nm. Fluid phase: 0.5 M H_3_PO_4_–triethylamine (pH = 2.4, phase A): acetonitrile (phase B) = 82:18. The concentration of ENR and CIP was determined using a standard curve. The standard curve was from 0.02 to 5.0 μg/mL, and both the correlation coefficient (r) for the ENR and CIP standard curve was 1.0. Both the limits of detection (LOD) and quantitation (LOQ) were 0.02 μg/mL. The recovery rates of three different added concentrations in tissues, for ENR and CIP, were 79.25–98.49% and 68.94–98.64%, respectively, and the RSD of precision was less than 11%.

### 4.4. Cell Culture and Liver Cytotoxicity Assay

According to Chen et al. [26], about 4% ENR would be metabolized into CIP in the pig liver; therefore, the cytotoxicity of the ENR granules to the pig liver was investigated by incubating hepatocytes with ENR and CIP, respectively. The pig hepatocytes were isolated from the pig livers in the control group. After the pig livers of the control group were sampled for the residue experiment, the rest of the livers were treated to make them into hepatocytes. The preparation processes of the pig hepatocytes are introduced in detail in the Appendix A. The obtained pig hepatocytes were cultured using a previously described method [27]. Briefly, the hepatocytes were cultured by DMEM medium that contained 10% fetal calf serum, 100 U/mL penicillin, and 100 U/mL streptomycin, and with an atmosphere of 5% CO_2_ at 27 °C. Cells were seeded into 96-well plates at concentrations of ~1 × 10^5^ cells/mL. After being cultured by DMEM medium for 4 h, the medium was replaced by the exposure medium that contained different concentrations of ENR or CIP, ranging from 10 to 1000 µg/mL (final concentration of cell medium), which was prepared by ENR and CIP standard. Briefly, the 10.1 mg ENR and 11.8 mg CIP were weighed in a 1.5 mL sterilization tube, respectively, and the 1 mL bacteria-free methanol was added to prepare a drug stock solution. The stock solution was diluted into an exposure medium with a different concentration range (10–1000 µg/mL), by DMEM cell medium.

After the cells were incubated with an exposure medium for 24 h, the 100 μL CCK-8 solution was added and cultured for an additional 2 h. After incubation, the absorbance of each group was detected at 450 nm using a microplate reader (multimode plate reader, Envision, Fremont, USA). Each concentration was performed in 3 different batches. The control groups were treated with different concentrations of methanol, which had the same dilution ratio as each drug-treated group. The blank group only contained the CCK-8 solution and DMEM medium. According to the absorbance, the cell viability was calculated using the following equation:$$\mathrm{Relative}\mathrm{cell}\mathrm{viability}=\frac{A-C}{B-C}\times 100\%$$
where *A* is the mean absorbance of each drug-treated group, *B* is the mean absorbance of each control group, and *C* is the mean absorbance of the blank group.

### 4.5. Construction of the PBPK Model

The PBPK model of ENR granules in the pigs was built by acslXtreme (version 3.0, the AEgis, Technologies Group, Inc. Huntsville, AL, USA). To describe the in vivo processes of ENR, the model includes the parent drug (ENR) module and metabolite (CIP) sub-module. The ENR module includes the oral administration module, gastrointestinal, plasma, lung, muscle, fat, liver, kidney, and rest of body, and the CIP sub-module includes the plasma, lung, muscle, fat, liver, kidney, and rest of body. The ENR was metabolized into CIP in the liver, then the CIP entered into the CIP sub-module and reached tissues, via the blood flow (Figure 6) [28]. Due to the compartment linking function, the blood compartment needs to be included. The muscle, fat, liver, and kidney were modeled as individual compartments, since they are interesting compartments in this study. Meanwhile, the liver and kidney are also responsible for metabolism and excretion. The lung was considered as one compartment, because the ENR is also used to treat pneumonic infections (e.g., *Mycoplasma* pneumonia, and infectious pleuropneumonia). All the compartments were assumed to be blood flow-limited and well stirred. The chemical-specific parameters for ENR in pigs were from the literature, and are provided in Appendix A, respectively [29,30,31,32]. Equations and the complete modeling code describing the administration, absorption, distribution, metabolism, and elimination processes of ENR are provided and clearly explained in the Appendix A. The oral PBPK model of ENR in pigs was well introduced by Lin et al.; the modeling code in our paper was cited from their paper [28]. Meanwhile, the molding code can also be found on the website of the College of Veterinary Medicine, Kansas State University (http://iccm.k-state.edu/, accessed on 15 June 2016). Additionally, in view of the PK difference between different ENR products, the visually reasonable values for KurineC1, Ka, and the partition coefficients (PCs) were obtained by an iterative manual adjustment approach to fit the experimental data for model calibration in the present study. The predictive power of the built PBPK model was verified by the observed plasma data in our previous studies (Appendix A) [3], the concentration of ENR and CIP in tissues were predicted directly by the PBPK model in the present study.

The residual concentration of ENR, CIP, and ENR plus CIP in edible tissues was used as observed data to evaluate the PBPK model by linear regression analysis with the predicted residue data. The R^2^ values were calculated and the model simulation was considered to be acceptable if the R^2^ value was ≥0.75 [15,17,21]. After the predictive accuracy of drug concentration in the tissue was verified, the oral dose that would cause toxicity to in vivo hepatocytes was reversed, based on the viability inhibition concentration against hepatocytes in vitro.

### 4.6. Parameter Sensitivity Analysis

To obtain the parameters that would influence the WTs significantly, PSA was performed by the embedded PSA module in the acslXtreme software. Briefly, the influence of the modeling parameters on the concentration of ENR plus CIP at 228 h (final running time of model) in the muscle, fat, liver, and kidney tissues was examined. On the other hand, after the sensitive parameters were obtained, the effects of different disease statuses on the WT could be analyzed. The sensitivity of the parameters was evaluated by the absolute value of NSC (/NSC/). The NSC was calculated using the equation below, which was reported previously [21,28].
$$\mathrm{NSC}=\frac{\Delta r}{r}\times \frac{p}{\Delta p}$$
where *r* is the model original output value from the original parameter value, Δ*r* is the change in model output, caused by a 1% increase of the original parameter value, *p* is the original parameter value, and Δ*p* is 1% of the original parameter values. The parameters were considered highly sensitive if /NSC/ ≥0.5 and medium sensitive if 0.5 > /NSC/ ≥ 0.2 [17,33,34]. (“/NSC/” means the absolute value of NSC, please retain it as /NSC/.)

### 4.7. Monte Carlo Analysis

The intra-species variability will influence the withdrawal intervals of ENR granules in pigs, and MC analysis needs to be performed to ensure the pork products are safe enough for humans. The analysis method was described in detail previously [31,32]. Briefly, the sensitive parameters that are recognized in Section 4.6 were used for MC analysis. PCs, Kmc, frac, KurineC1, KurineC, and Kint were assumed as normal distributions, and their mean values were obtained by parameter optimization. The CVs of body weight (BW) were calculated by the actual BW of the experiment pigs. For parameters without measured values, a default value of 20% was used for PCs and 30% for other chemical-specific parameters [17,21]. Standard deviations (SD) of PCs and chemical-specific parameters were calculated by mean multiply CV. Therefore, the lower limit (mean–SD) and upper limit (mean + SD) are defined. Each MC simulation was set as a batch run of 1000 iterations in the MC module that was included in the acslXtreme software. For each simulation, the random numbers of these parameters, with the mean value, SD, lower bound, and upper bound, were firstly input into the acslXtreme software, and then they were incorporated into this multi-route model to predict the residue curves of ENR plus CIP in tissues of 1000 virtual individuals [11,35,36].

### 4.8. Withdrawal Time Prediction

As mentioned above, after each run, the predicted concentrations of ENR plus CIP versus time data were all automatically generated by the acslXtreme software. The MC analysis provided all the predicted concentrations in 1000 virtual individuals. The predictive WTs were determined as the times when total concentrations of ENR + CIP in the edible tissues fell below the MRLs for 99% of the simulated population [17,37]. According to China and the Codex Alimentarius Commission (CAC), the residual marker of ENR products was ENR plus CIP, and the MRLs of ENR plus CIP in the muscle, fat, liver, and kidney of the pig were 100 µg/kg, 100 µg/kg, 200 µg/kg, and 300 µg/kg, respectively [38]. The WTs distribution of the 1000 virtual individuals was counted based on the MC results, thus the predictive WTs that met 99% of the simulated population were obtained. Meanwhile, the predicted WTs were compared with the observed WTs, calculated by WT 1.4 software, to evaluate the applicability of the built PBPK model.

### 4.9. In Vivo Liver Toxicity Dose

According to the relative cell viability of cells versus concentration, the inhibition curves of ENR and CIP against pig liver cells were fitted by GraphPad Prism software (version 7.0, GraphPad Software Inc., La Jolla, CA, USA). The IC_50_ values of ENR and CIP were calculated by the inhibition curves. Then, we assumed that when the in vivo liver was exposed to this IC_50_, the ENR granules would show toxicity to the pig liver. Subsequently, the IC_50_ values of ENR and CIP were treated as in vivo peak concentrations (C_max_, toxicity threshold concentration) in liver tissue. Then, the C_max_ was reversed to an oral toxicity dose by the PBPK model, thus the oral toxicity dose of the ENR granules that could produce this C_max_ in pig liver was obtained. In other words, the toxicity dose of the prepared ENR granules to the liver was proposed by the PBPK model, rather than animal experiments. To directly observe the safe dose range, all the concentrations of the cell exposure medium were reversed to the related dose.

### 4.10. Statistical Analysis

Data are presented as mean ± standard deviation (SD). The one-way analysis of variance was adopted for statistical analysis. Statistical significance was defined at a *p*-value of 0.05 by the SPSS software (version 20, IBM, New York, NY, USA).

## 5. Conclusions

To promote the further development of the previously developed ENR granules, and its judicious use in the veterinary clinic, a PBPK model of ENR in pigs was built to determine the WT and evaluate the liver toxicity of the ENR granules. Compared with the measured residue concentrations, the concentrations of ENR, CIP, and ENR plus CIP, in edible tissues, were well predicted by the PBPK model (R^2^ > 0.82), suggesting that the drug concentration curves in the tissues at other doses could be extrapolated. To ensure the pork products were safe enough for humans, after the pigs were administrated ENR granules, the population WTs of the ENR granules in pigs were investigated using MC analysis. Moreover, the results indicated that the predictive WTs of the ENR granules in the muscle, fat, liver, and kidney were 5 d, 3 d, 6 d, and 4 d, respectively. Compared with the measured data, the final WT (6 d) of the ENR granule was well predicted by the PBPK model. Finally, the in vitro IC_50_ value (225.9 µg/mL) of ENR against pig hepatocytes was reversed to an oral toxicity dose by the PBPK model, which indicated that the liver would be exposed to the IC_50_ when the pigs were administrated a dose of 130 mg/kg b.w., suggesting that the ENR granules would not induce toxicity to the pig liver when the model was applied in the clinic. Overall, the WT and the liver toxicity dose of the prepared ENR granules to pigs were well predicted by the PBPK model in the present study. Our studies indicated that the PBPK model is a potential tool for the monitoring of food safety, especially in label-extra drug use, and is also an excellent tool for reducing the costs of the development of veterinary antibiotics. This is also the first time that the PBPK model has been applied to determine the toxicity dose of veterinary antibiotics. Our research will provide a new model for adopting an in silico method to develop new veterinary antibiotics.

## Figures and Tables

**Figure 1 antibiotics-10-00955-f001:**
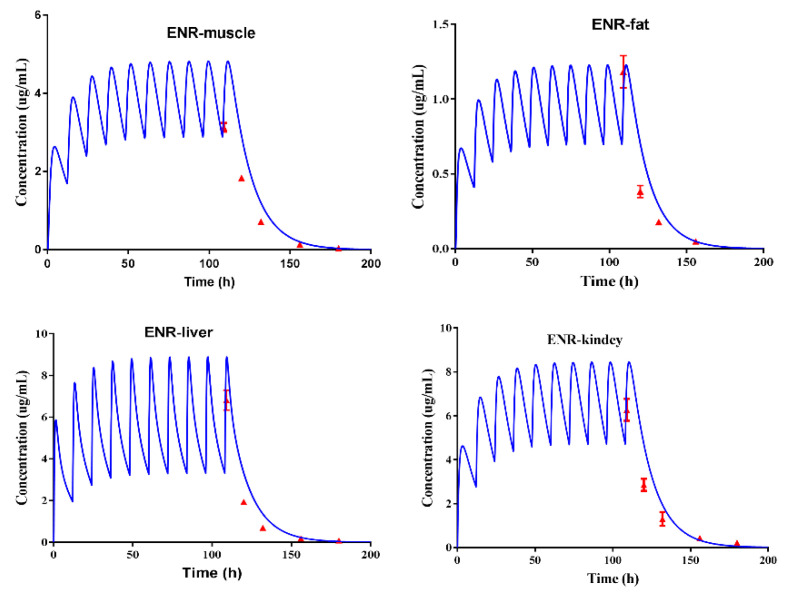
PBPK model calibration results for ENR tissue residues in the pig. Comparisons of model-simulated data (lines) and measured concentrations data (symbols) of ENR in the tissues of the pig after administrated at a dose of 5 mg/kg b.w. twice per day.

**Figure 2 antibiotics-10-00955-f002:**
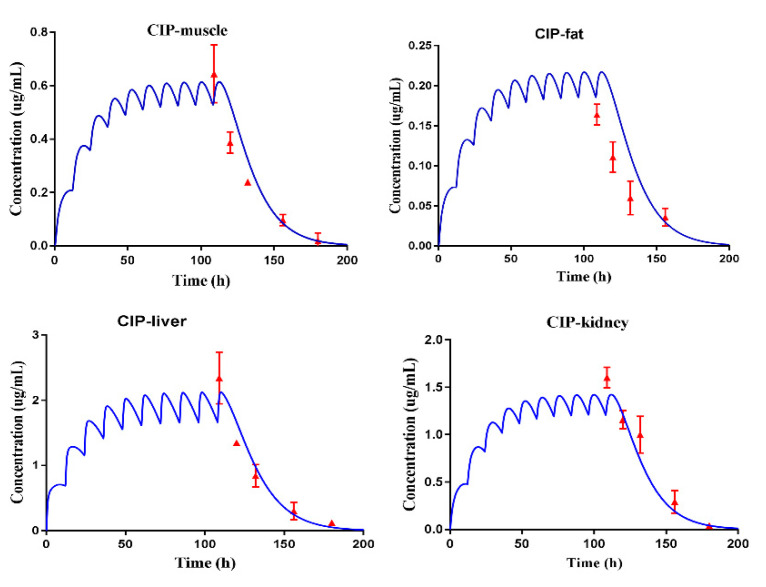
PBPK model calibration results for CIP tissue residues in the pig. Comparisons of model-simulated data (lines) and measured concentrations data (symbols) of CIP in the tissues of the pig after administrated at a dose of 5 mg/kg b.w. twice per day.

**Figure 3 antibiotics-10-00955-f003:**
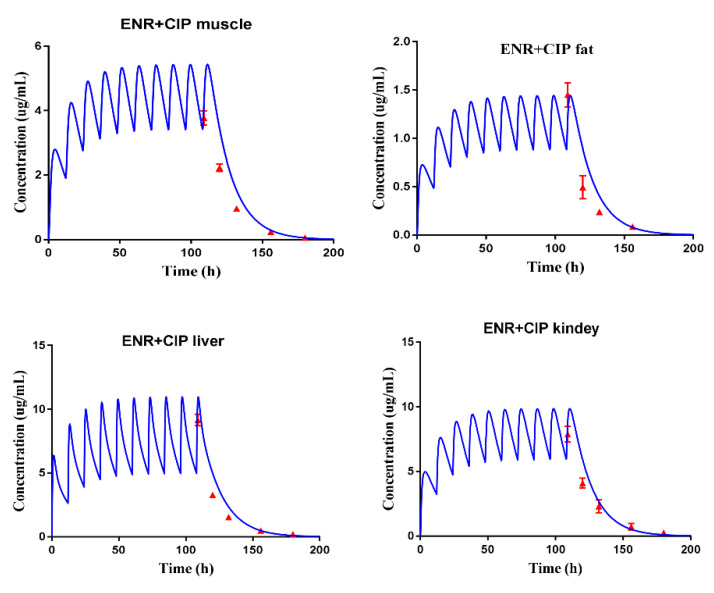
PBPK model calibration results for ENR plus CIP tissue residues in the pig. Comparisons of model-simulated data (lines) and measured concentrations data (symbols) of ENR plus CIP in the tissues of the pig after administrated at a dose of 5 mg/kg b.w. twice per day.

**Figure 4 antibiotics-10-00955-f004:**
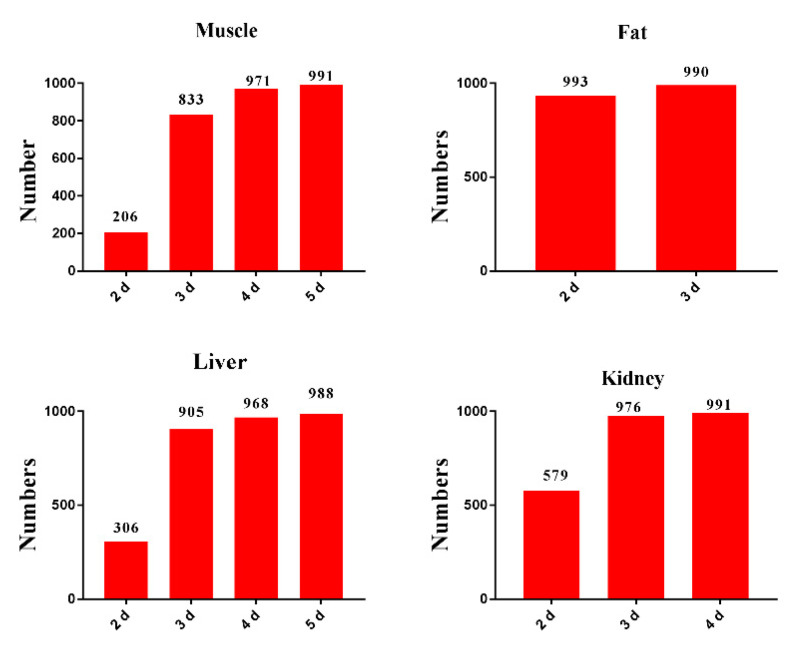
Cumulative distributions of withdrawal time after an oral administration at a dose of 5 mg/kg b.w. twice per daily for 5 days based on the Monte Carlo analysis. The predicted WT of ENR granules in muscle, fat, liver, and kidney of pigs was 5 d, 3 d, 6 d, and 4 d, respectively.

**Figure 5 antibiotics-10-00955-f005:**
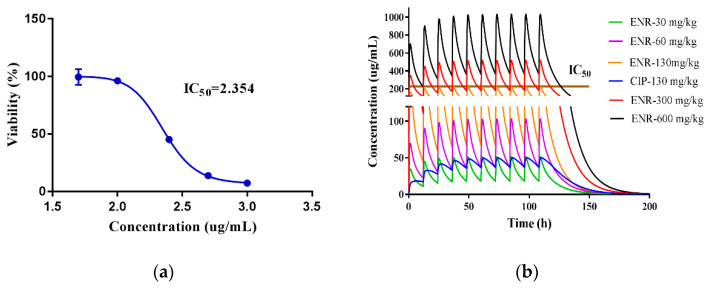
The IC_50_ value of ENR against pig hepatocytes (**a**) and the concentration of ENR and CIP in the liver at different doses for 5 days (**b**). The log IC_50_ value of ENR against pig hepatocytes was 2.354 mg/kg b.w. and the in vivo liver toxicity dose of ENR granules was 130 mg/kg b.w.

**Figure 6 antibiotics-10-00955-f006:**
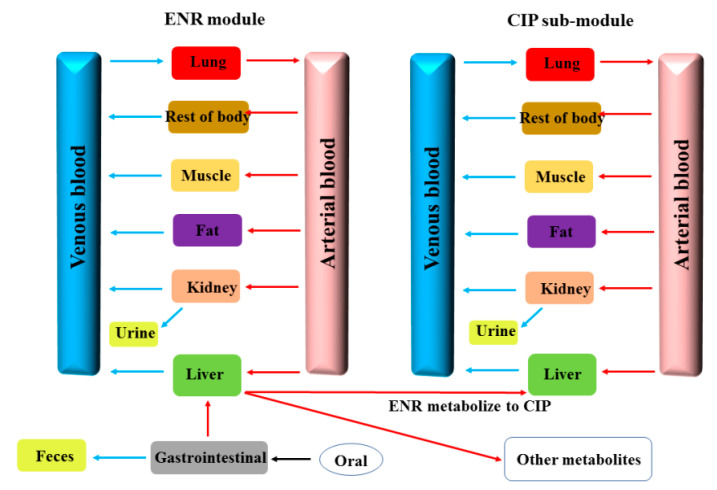
The schematic of the PBPK model for ENR granules in the pig.

## Data Availability

Not applicable.

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
