# Peer review of "Apply a Physiologically Based Pharmacokinetic Model to Promote the Development of Enrofloxacin Granules: Predict Withdrawal Interval and Toxicity Dose"

_antibiotics, 2021, doi:10.3390/antibiotics10080955_

Round 1

Reviewer 1 Report

While this is a very interesting paper with very nice results, namely the use of in silico PBPK methods in WT estimation and toxicity testing and thereby limiting the use of  animal trials , I cannot accept the paper in its current form due to many English spelling and grammar mistakes. These severely hamper the readability of the manuscript. I started correcting some of the spelling mistakes myself, but it is simply too much work. Clear and correct communication is essential in science. Therefore, I suggest the authors contact an expert in English to proofread and correct their manuscript.

Major comments:

The authors should mention the critical status of fluoroquinolones in human medicine (WHO) and that use of fluoroquinolones is strictly controlled is veterinary medicine. Is there any legislation in China for controlling fluoroquinolone use? Also make clear that this paper helps in curbing  antimicrobial resistance from a One Health perspective, i.e. limiting residues in pigs and therefore limit resistance in humans. Mention this in the introduction.

With regard to the results, the authors state minor differences between model estimated and true WTs. However, sometimes there are differences of at least two days in some tissues. Noted, the overall WT is the same, but still. These discrepancies and a possible explanation for them should be discussed.

Other minor comments regarding the paper:

ABSTRACT:

Overall abstract: many spelling and grammar mistakes 

Line 24: was reversed? What do the authors mean by this?

Line 26: promoting antibiotics rational use à e.g. better writing: promote judicious use of antimicrobial agents

Line 26: veterinary antibiotic’s products is not correct English à veterinary antimicrobial products

INTRODUCTION

Line 32: skip "the", just intestinal bacterial infections

Line 34: losses to the farming is not correct English

Line 36: a new ENR formulation was developed in previous studies. Could you briefly explain what is in the formulation? This will be of interest for the audience. How are these administered? Via feed? Additionally, why were these developed? What is the advantage versus parental administration?

Moreover, will this new formulation be commercially exploited? Will this pass the regulatory agencies?

Line 37: How was the dose of 5 mg/kg proposed? Explain this.

Lines 30-43 are very badly written. I corrected some the English mistakes throughout the manuscript but not all, because there are too many. In the next revision, see that an English expert is consulted and that the manuscript has proper spelling and grammar. This will make the manuscript much more attractive to the audience.

Line 45 – 46: WT needs to be determined in order to have a successful antibiotic? Not because of residue and food safety issues? The following sentences (lines 46 – 48 belong to sentence line 45).

Line 52: broad-spectrum is outdated terminology. State against which bacteria the drug works (Gram negative or postitive, aerobic or anaerobic)

Line 52 – 55: too many and… and….and

What is meant by line 56-58? Extra-label use of fluoroquinolones? Is this allowed?

Line 60: What about Baytril® 10% oral solution ?

Line 63: Acibenser baerii --> Latin names in italic

Lines 64-65: write in full

Line 65: with regard to liver toxicity, say something about the clearance / elimination of enrofloxacin in pigs, e.g. renal and extra-renal processes.

Line 71: VICH, explain abbreviation. 

Line 71: many animals "are" needed

Line 73-74: why are 16 and 32 pigs needed according to VICH, please briefly explain.

Line 76: why will this limit food safety estimation? These tests for WT are mandatory in order to go to market.

Line 87: skip “more”. This paragraph was well written.

Line 89 :”needs”

Line 93: by combining with not proper English à were predicted using MC

Line 95: reversing?

RESULTS

Lines 110 – 111: sentence does not make sense

Line 111: Don’t start sentence with “And”, “liner” regression?

Line 123: Why is there a figure with both ENR and CIP? The results are given in previous two figures and therefore, Figure 3 seems to be redundant.

Line 156: don’t start with “And”

Line 167-168: sentence grammatically incorrect. Additionally, what is meant with CIP is produced ENR via liver, in the experiments? Or in vivo? If the latter, what % of ENR is converted to CIP?

Lines 170 – 171: I don’t understand what the authors mean.

Line 175: why the assumption of 75%? Please explain.

Line 185: What is meant by the negative inhibition rate?

DISCUSSION

Lines 191 – 203:  very badly written

Line 199: reminded is not the right word, better: showed

Line 213: reversing?

Line 232: add reference

Line 278: What is meant with “resistance” in this context?

Overall: Discussion part many spelling / grammar mistakes…

MATERIALS AND METHODS

Line 314: Animal trials?

Part 4.2 and 4.3 can be merged together In my opinion

Line 316: were the pigs fed at libitum? What about water intake?

Line 319: Could the authors also provide a reference with the optimal stable conditions for pigs?

Line 336: EU Directive should be quoted as 2010/63/EU

Line 351 and throughout the rest of the Materials and Methods, when describing the instruments, e.g. HPLC column, next to the country of the instruments, also mention the city

Line 366: How much of ENR is metabolized to CIP? Add reference

Line 467: WT = 99 % of the population below the MRL, but what about the 95% CI?

Author Response

Reviewer#1:

Comment: While this is a very interesting paper with very nice results, namely the use of in silico PBPK methods in WT estimation and toxicity testing and thereby limiting the use of animal trials, I cannot accept the paper in its current form due to many English spelling and grammar mistakes. These severely hamper the readability of the manuscript. I started correcting some of the spelling mistakes myself, but it is simply too much work. Clear and correct communication is essential in science. Therefore, I suggest the authors contact an expert in English to proofread and correct their manuscript.

Response: Thank you for your comments and your kindness. The English spelling and grammar were checked carefully. I believe that the manuscript was easy to read and understand currently.

Major comments:

Comment: The authors should mention the critical status of fluoroquinolones in human medicine (WHO) and that use of fluoroquinolones is strictly controlled is veterinary medicine. Is there any legislation in China for controlling fluoroquinolone use? Also make clear that this paper helps in curbing antimicrobial resistance from a One Health perspective, i.e. limiting residues in pigs and therefore limit resistance in humans. Mention this in the introduction.

Response: Thank you for your comments. Actually, as what you suggested, limiting residues in food-producing animals therefore limit resistance in humans is one of the initiatives of the current manuscript. The critical status of fluoroquinolones in human medicine and veterinary medicine were highlighted in the Instruction section Meanwhile, the action of limiting residues in pigs and therefore limits resistance in humans was mentioned in the Instruction section (line 61-66).

Comment: With regard to the results, the authors state minor differences between model estimated and true WTs. However, sometimes there are differences of at least two days in some tissues. Noted, the overall WT is the same, but still. These discrepancies and a possible explanation for them should be discussed.

Response: Thank you for your comments. These discrepancies and a possible explanation of the differences WTs between prediction and true were further discussed (line 261-265).

Other minor comments regarding the paper:

ABSTRACT:

Comments: Overall abstract: many spelling and grammar mistakes.

Line 24: was reversed? What do the authors mean by this?

Response: It means that the dose of ENR granules that could produce the cytotoxicity concentration in pig liver was calculated by PBPK model based on the cytotoxicity concentration (225.9 µg/mL) of ENR against pig hepatocytes (Figure 5). The “reversed” was changed into “calculated” (line 24).

Comment: Line 26: promoting antibiotics rational use à e.g. better writing: promote judicious use of antimicrobial agents

Response: Thank you for your comments. The related description was changed (line 26, line 222, and line 524).

Comment: Line 26: veterinary antibiotic’s products is not correct English à veterinary antimicrobial products

Response: It was reversed (line 27).

INTRODUCTION

Comment: Line 32: skip "the", just intestinal bacterial infections

Response: It was skipped.

Comment: Line 34: losses to the farming is not correct English

Response: The “farming” was changed into “livestock and poultry breeding” (line 34).

Comment: Line 36: a new ENR formulation was developed in previous studies. Could you briefly explain what is in the formulation? This will be of interest for the audience. How are these administered? Via feed? Additionally, why were these developed? What is the advantage versus parental administration? Moreover, will this new formulation be commercially exploited? Will this pass the regulatory agencies?

Response: Thank you for your comments. Actually, as what you suggested, we should have to provide more details about the new ENR formulation. Considered the length of Instruction, the details were not provided in the previous manuscript.

Briefly, to improve the palatability of ENR and provide an oral ENR product that can be group administered, an ENR granule that could be administrated by mixing with feed was prepared. The ENR granule was prepared by solid dispersion technology, and starch as well as sodium chloride was used as excipient. According to Animal Drug @FDA (https://animaldrugsatfda.fda.gov/adafda/views/#/search), there is only injection for pigs on the market currently. As we known, the injection is not convenient for large scale livestock and poultry breeding. In view of there is no an ENR oral product for pigs on the market currently, we are going to apply for the new drug certificate and production approval from Ministry of agriculture of China, thus to promote the commercialization of the developed ENR granule. As for if this application will pass the regulatory agencies, I am afraid of that I cannot answer you directly. After all, the new drug review is a long process. Related descriptions were added in the revised manuscript (line 37-39).

Comment: Line 37: How was the dose of 5 mg/kg proposed? Explain this.

Response: As shown in Table S1, when the dose was 5 mg/kg, the ratio of AUC24h/MIC90 (AUCI) was 133. As we known, for a fluoroquinolone antibiotic, when the AUC24h/MIC90 is above 120, its good antibacterial ability against gram-negative bacteria can be expected (  Schuck et al. Pharmacokinetic/pharmacodynamic (PK/PD) evaluation of a once-daily treatment using ciprofloxacin in an extended-release dose form. Infection 2005, 33, 22–28, doi:10.1007/s15010-005-8204-0. Craig, W.A. Does the dose matter? Clin. Infect. Dis. 2001, 33, Suppl 3, S233–S237, doi:10.1086/321854). Therefore, to prevent and control the infections caused by Salmonella and E. coli, the dose of 5 mg/kg was proposed. The detail was added in the revised Supplementary Information Table S2.

Comment: Lines 30-43 are very badly written. I corrected some the English mistakes throughout the manuscript but not all, because there are too many. In the next revision, see that an English expert is consulted and that the manuscript has proper spelling and grammar. This will make the manuscript much more attractive to the audience.

Response: Thank you for your kindness. The spelling and grammar were checked carefully.

Comment: Line 45-46: WT needs to be determined in order to have a successful antibiotic? Not because of residue and food safety issues? The following sentences (lines 46 – 48 belong to sentence line 45).

Response: Thank you for your comments. Actually, before a new veterinary antibiotic product on the market, if it was used for food-producing animals, the WT data was required by the regulatory authorities. Meanwhile, the requirement also is for food safety consideration. Only followed the WT principle, the veterinary antibiotic products can be used judiciously in the veterinary clinic.

Comment: Line 52: broad-spectrum is outdated terminology. State against which bacteria the drug works (Gram negative or positive, aerobic or anaerobic)

Response: Thank you for your comments. The related description was revised (line 58-59).

Comment: Line 52 – 55: too many and… and….and

Response: The sentence was re-written.

Comment: What is meant by line 56-58? Extra-label use of fluoroquinolones? Is this allowed?

Response: Thank you for your comments. In fact, although the extra-label use is not allowed. However, due to lack of professional knowledge, to make the sick animals recover as soon as possible, the farmers may use antibiotics over dose. On the other hand, there is no special drug for some food animals (e.g., rabbit, deer, and duck) in China, when these animals are infected, to save their life and prevent the spread of disease, we could not forbid antibiotics use. These extra-label use seems not reasonable. But this is the gap between science and reality. Therefore, we propose that apply the PBPK model to predict the residue, thus to ensure the food safety as much as possible.

Comment: Line 60: What about Baytril® 10% oral solution?

Response: Thank you for your comment. In fact, Baytril® 10% oral solution is used for poultry. As we known, the taste of poultry is poor than pig. Therefore, the ENR oral solution can be used for poultry. Currently, the ENR products for pig use is injection only (FDA, https://animaldrugsatfda.fda.gov/adafda/views/#/search).

Comment: Line 63: Acibenser baerii --> Latin names in italic

Response: It was revised (line 73)

Comment: Lines 64-65: write in full

Response: It was revised (line 74-75).

Comment: Line 65: with regard to liver toxicity, say something about the clearance / elimination of enrofloxacin in pigs, e.g. renal and extra-renal processes.

Response: Thank you for your comments. The related information was added in the revised manuscript. “Meanwhile, the conclusion meets the report of Food and Agriculture Organization (FAO) that the highest ENR concentration initially occurred in the liver in all species. And in all species studied, elimination was primarily via the urine and feces (FAO, http://inchem.org/documents/jecfa/jecmono/v34je05.htm)” (line 79-83).

Comment: Line 71: VICH, explain abbreviation.

Response: The abbreviation was explained (line 86-88).

Comment: Line 71: many animals "are" needed

Response: The mistake was revised (line 88).

Comment: Line 73-74: why are 16 and 32 pigs needed according to VICH, please briefly explain.

Response: According to VICH, for the determination of WT in pigs, four sampling timepoints are needed, and per timepoint should include four pigs at least (https://www.fda.gov/media/78351/download, page 6). Therefore, 16 pigs are needed for WT determination. For the target-species safety, the animal needs to be divided into control group, low dose group, medium dose group, and high dose group. And eight pigs are needed for each group. Therefore 32 pigs are needed for the target-species safety.

Comment: Line 76: why will this limit food safety estimation? These tests for WT are mandatory in order to go to market.

Response: Thank you for your comments. As mentioned above, the large number of animals are needed for WT determination. The large number of animal uses will cause a huge economic burden to drug companies and farmers. Especially, for the big animal (e.g., pig, cattle, and sheep), they are expensive. To reduce costs, drug companies and farmers may not slaughter so many pigs, cattle or sheep, thus inadequate data will induce the food safety worries. Therefore, the silico replace tool is needed.

Comment: Line 87: skip “more”. This paragraph was well written.

Response: It was deleted (line 104).

Comment: Line 89: “needs”

Response: It was revised (line 106).

Comment: Line 93: by combining with not proper English à were predicted using MC

Response: It was revised (line 110).

Comment: Line 95: reversing?

Response: The sentence was re-written.

RESULTS:

Comment: Lines 110–111: sentence does not make sense

Response: The sentence was simplified (line 128).

Comment: Line 111: Don’t start sentence with “And”, “liner” regression?

Response: Thank you for your comments. The “And” was deleted. The “liner” was changed into “linear”.

Comment: Line 123: Why is there a figure with both ENR and CIP? The results are given in previous two figures and therefore, Figure 3 seems to be redundant.

Response: Thank you for your comments. Actually, due to the CIP (the main metabolite of ENR) possesses the antibacterial activity. Therefore, the residual marker of ENR is ENR plus CIP (VICH). Although the predicted concentration of ENR and CIP both were well fitted by the measured data, to show the residual curve of residual marker (ENR plus CIP) clearly, the Figure 3 was drawn.

Comment: Line 156: don’t start with “And”

Response: It was deleted.

Comment: Line 167-168: sentence grammatically incorrect. Additionally, what is meant with CIP is produced ENR via liver, in the experiments? Or in vivo? If the latter, what % of ENR is converted to CIP?

Response: Thank you for your comments. Actually, ENR can produce CIP through the liver metabolizing in vivo. According to FAO, the metabolism rate is different in different animals. In swine about 4% ENR will be metabolized into CIP (Chen et al., 2006, Comparison of enrofloxacin excretion between oral and intramuscular administration in swine). In rats, 26-31% ENR will be metabolized into CIP. In houses, the metabolism rate is 20-35%. As for goats, the metabolic conversion is about 36% (FAO, http://inchem.org/documents/jecfa/jecmono/v34je05.htm).

Comment: Lines 170-171: I don’t understand what the authors mean.

Response: I am sorry for the confusion. It meant that the in vivo liver toxicity of ENR granules was mainly caused by ENR rather than CIP. The sentence was re-written (line 188).

Comment: Line 175: why the assumption of 75%? Please explain.

Response: Thank you for your comments. Actually, as mentioned at line 194 “We assumed that when the concentration of ENR in the in vivo liver reached this IC50 value”, the assumption was 50% viability of pig hepatocytes.

Comment: Line 185: What is meant by the negative inhibition rate?

Response: When the dose of ENR granule was 130 mg/kg b.w., the Cmax of ENR in liver was 222.9 µg/mL and that of CIP was 51.9 µg/mL. As mentioned as Table S6, when the concentration of CIP was 51.9 µg/mL, the viability of pig hepatocytes was 100%. In another word, when the dose of ENR granule was 130 mg/kg b.w., the metabolized CIP would not induce hepatocytes toxicity. The “negative inhibition rate” was changed into “low inhibition rate”.

DISCUSSION

Comment: Lines 191-203: very badly written

Response: The sentences were checked and re-written carefully (line 210-223).

Comment: Line 199: reminded is not the right word, better: showed

Response: It was revised (line 219).

Comment: Line 213: reversing?

Response: It was changed into “predicting” (line 224).

Comment: Line 232: add reference

Response: The related reference was added (line 253).

Comment: Line 278: What is meant with “resistance” in this context?

Response: It means that the kidney cells were more tolerance to ENR. The “resistance” was changed into “tolerance” (line 303).

Comment: Overall: Discussion part many spelling / grammars mistakes…

Response: Thank you for your guidelines. The spelling and grammars of whole manuscript were checked carefully.

MATERIALS AND METHODS

Comment: Line 314: Animal trials?

Response: Actually, section 4.2 provided the information of animals, including feed condition, and animal welfare. The animal trials were performed at section 4.3.

Comment: Part 4.2 and 4.3 can be merged together in my opinion.

Response: Thank you for your comments. We also think that if Part 4.2 and 4.3 are merged together, the manuscript will easy to be understood. However, generally speaking, the animal welfare and feed condition are quite important for animal trials. Therefore, we provided the animal information individually.

Comment: Line 316: were the pigs fed at libitum? What about water intake?

Response: Actually, as mentioned as line 330, the pigs were fed with drinking freely. The pigsty is equipped drinking water device; thus, the pigs can drink water freely.

Comment: Line 319: Could the authors also provide a reference with the optimal stable conditions for pigs?

Response: Thank you for your comments. The optimal stable conditions for pigs are cited form Guide for the Care and Use of Laboratory Animals (National Research Council (US) Committee). The reference was added in the revised manuscript (line 343).

Comment: Line 336: EU Directive should be quoted as 2010/63/EU

Response: It was revised (line 363).

Comment: Line 351 and throughout the rest of the Materials and Methods, when describing the instruments, e.g. HPLC column, next to the country of the instruments, also mention the city

Response: Thank you for your comments. The details were added.

Comment: Line 366: How much of ENR is metabolized to CIP? Add reference

Response: According to Chen et al (2006) about 4% ENR would be metabolized into CIP in pig liver (line 394). (Chen, et al. Comparison of enrofloxacin excretion between oral and intramuscular administration in swine. Veterinary Science in China, 2006, 36(7): 583-586. Web of Science).

Comment: Line 467: WT=99 % of the population below the MRL, but what about the 95% CI?

Response: Thank you for your comment. Actually, there are two common methods to calculate WT by PBPK model based on the function of the adopted software: 1. Calculating by 99 % of the population below the MRL. 2. Calculating by 99 % of the population below the MRL with 95% CI. For instance, Xu et al. (2020) calculated the WT of doxycycline in grass carp by 99 % of the population below the MRL with 95% CI by Berkeley Madonna (https://doi.org/10.1016/j.fct.2020.111127). Yang et al (2019). calculated the WT of florfenicol in cattle by 99 % of the population below the MRL by acslXtreme (https://doi.org/10.1016/j.fct.2019.02.029).

As what you suggested, in view of the higher prediction accuracy, we should have to adopt 95% CI method. Unfortunately, we do not possess the right of Berkeley Madonna software. Besides, 99 % of the population below the MRL is enough from a statistical point of view. After all, according to the method used in China and EU, the WT was finally calculated to ensure that the concentrations of total residues of ENR plus CIP in 95 % of the population were below the MRL. On the other hand, the above two publications both were published by Food and Chemical Toxicology. Therefore, we believe that calculate the WT by 99 % of the population below the MRL can be accepted. The comments from you are significant helpful for us to improve the manuscript. Thank you for your guidelines and patience sincerely.

Reviewer 2 Report

The presented study is of huge importance in animal production and to guarantee that the withdrawals periods are established and consumers are protected. For all veterinary medicines those studies should be performed and new approaches compared with conventional ones. Apart from some English improvements, there are some comments to be addressed:

The number of animals used for this study (18 pigs in the experimental group) is considered statistical enough to achieve consistent results?

Line 46: Rephrase in order to avoid starting a sentence with “Because”. The same for line 194: “And based…”

MRL – should be presented in the first appearance: in the line 147 “... below tissue MRLs at different withdrawal...” and after in the line 217-218 “maximum residue limit 217 (MRL),” Also add the reference of the MRLs considered in the present study at the first time it is mentioned. Add the same reference in table S2.

Authors should provide information about the LOD and LOQ of the method. Some results are presented in table S2 as below LOD but no information regarding those analytical limits is available.

Author Response

Reviewer#2

The presented study is of huge importance in animal production and to guarantee that the withdrawals periods are established and consumers are protected. For all veterinary medicines those studies should be performed and new approaches compared with conventional ones. Apart from some English improvements, there are some comments to be addressed:

Comment: The number of animals used for this study (18 pigs in the experimental group) is considered statistical enough to achieve consistent results?

Response: Thank you for your comments. Actually, for this study, the measured residual data was used for evaluate the predictive accuracy of the PBPK model. According to previous studies, for the model evaluation, four measured value can be accepted (Yang et al., 2019, https://doi.org/10.1016/j.fct.2019.02.029). Even one measured value can also be accepted (Lin et al., 2016). To predict the WT and toxicity dose with higher confidence, more animals were adopted in our present studies. On the other hand, according to VICH, four pigs per timepoint is enough for a complete WT experiment. Therefore, we believe that 18 pigs in the experimental group (three pigs in each timepoint) was statistical enough to achieve reasonable model evaluation.

Comment: Line 46: Rephrase in order to avoid starting a sentence with “Because”. The same for line 194: “And based…”

Response: Thank you for your comments. The related descriptions were revised (line 48 and line 214).

Comment: MRL–should be presented in the first appearance: in the line 147 “... below tissue MRLs at different withdrawal...” and after in the line 217-218 “maximum residue limit 217 (MRL),” Also add the reference of the MRLs considered in the present study at the first time it is mentioned. Add the same reference in table S2.

Response: Thank you for your comments. The abbreviation was checked in the revised manuscript (line 164). Also, actually, the reference of the MRLs considered was mentioned in line 494-497. Due to the result section is arranged in the front of method section, thus the reference may be not so clear. We should have to put the reference of the MRLs considered in the present study at the first time it is mentioned. However, it seems that put the reference of the MRLs in the result section is not so reasonable. Therefore, reference of the MRLs was put in line 494-497 in the revised manuscript. Additionally, to let the manuscript is easier to be understood, the reference was added in the Table S2.

Comment: Authors should provide information about the LOD and LOQ of the method. Some results are presented in table S2 as below LOD but no information regarding those analytical limits is available.

Response: Thank you for your comments. Actually, the LOD and LOQ were provided in line 389. Also, to make the manuscript easy to be read and understood, the values of LOD and LOQ were added to the Table S2. The comments from you are significant helpful for us to improve the manuscript. Thank you for your guidelines and patience sincerely.
